# Preparation, Properties, and Mechanism of Flame-Retardant Poly(vinyl alcohol) Aerogels Based on the Multi-Directional Freezing Method

**DOI:** 10.3390/ijms232415919

**Published:** 2022-12-14

**Authors:** Jixuan Wei, Chunxia Zhao, Zhaorun Hou, Yuntao Li, Hui Li, Dong Xiang, Yuanpeng Wu, Yusheng Que

**Affiliations:** 1School of New Energy and Materials, Southwest Petroleum University, Chengdu 610500, China; 2State Key Laboratory Oil and Gas Reservoir Geology and Exploitation, Chengdu 610500, China; 3The Center of Functional Materials for Working Fluids of Oil and Gas Field, School of New Energy and Materials, Southwest Petroleum University, Chengdu 610500, China

**Keywords:** poly(vinyl alcohol) composite aerogel, *α*-zirconium phosphate, phosphate cellulose, flame retardancy, multi-directional freezing

## Abstract

In this work, exfoliated *α*-zirconium phosphate (*α*-ZrP) and phosphated cellulose (PCF) were employed to synthesize poly(vinyl alcohol) composite aerogels (PVA/PCF/*α*-ZrP) with excellent flame retardancy through the multi-directional freezing method. The peak heat release rate (PHRR), total smoke release (TSR), and CO production (COP) of the (PVA/PCF_10_/*α*-ZrP_10_-3) composite aerogel were considerably decreased by 42.3%, 41.4%, and 34.7%, as compared to the pure PVA aerogel, respectively. Simultaneously, the limiting oxygen index (LOI) value was improved from 18.1% to 28.4%. The mechanistic study of flame retardancy showed evidence that PCF and *α*-ZrP promoted the crosslinking and carbonization of PVA chains to form a barrier, which not only served as insulation between the material and the air, but also significantly reduced the emissions of combustible toxic gases (CO_2_, CO). In addition, the multi-directional freezing method further improved the catalytic carbonization process. This mutually advantageous strategy offers a new strategy for the preparation of composite aerogels with enhanced fire resistance.

## 1. Introduction

The synthesis of resorcinol-formaldehyde polymer aerogels in the late 1980s by Pekala resulted in an unparalleled revolution in the field of materials [1,2]. Polymeric aerogels of poly(vinyl alcohol) (PVA) [3], polyimide (PI) [4], cellulose [5], pectin [6], etc., gained a lot of attention, due to their controllable structural diversity and desired mechanical properties. Compared to other polymer aerogels, the polar groups (-OH-) in PVA enable its easy dissolution in water. Moreover, PVA aerogels are the most widely investigated three-dimensional materials, due to their abundant source, non-toxicity, low cost, easy synthesis, and excellent biocompatibility [3,7,8]. However, due to lack of flame retardancy, applications of PVA have been seriously restricted in fields of aerospace, synthetic catalysis, chemical energy industry, biomedicine, intelligent manufacturing, etc. Many efforts have been made to develop flame retardant aerogels, mainly PVA/MMT aerogels [9,10], PVA/clay aerogels [11], PVA/alginate aerogels [12], PVA/h-BN@PDA@TiO_2_ aerogels [13], etc.

Two-dimensional (2D) layered nanomaterials have become hotspots of research, since they can be exfoliated into a single-layer or few-layered structures by atoms, molecules, small organic groups, and even polymers, bringing about significant changes in their properties and notable improvement of their dispersion in water [14,15,16,17]. Among them, layer structured *α*-zirconium phosphate (*α*-ZrP) has gained more interest, due to its chemical stability against strong acids/bases and redox agents. Moreover, it improves the chemical, mechanical, and thermal properties of the polymeric matrix. The proton of P-OH group in *α*-ZrP can be exchanged; according to the “acid-base proton theory”, *α*-ZrP is a Brønsted acid that can be used as a solid acid catalyst. At high temperature combustion, it can catalyze polymer crosslinking into carbon, forming a “barrier”, blocking the transmission of flammable gas, oxygen, and heat; it is a new type of efficient nano flame retardant. However, it is difficult to meet the flame retardant requirements of polymer when used alone. Therefore, it is often used to enhance the flame retardant property of polymer by synergistic flame retardant with other flame retardants [18]. Water molecules present in the cavity of the crystal structure form a hydrogen bond with P-OH present in the layer. By means of inhaling, embedding, inserting, pillaring, or grafting of organic groups of specific sizes, the interlayer spacing can be altered, which is an effect of van der Waals forces. This makes it an excellent matrix for the preparation of intercalated compounds and polymer/inorganic nanocomposites. The work of Lu [19] showed that the dispersion state, the chemical structure, and the slice state of *α*-ZrP significantly influence the flammability performance and thermal stability of the polymer materials. Therefore, commercial polyoxyalkylene amine (M1000) was used to assist the ultrasonic stripping of exfoliated layered *α*-ZrP into monolayers of *α*-ZrP nano-plates and obtain stable dispersed *α*-ZrP nano-plates in the present study. Since M1000 is solid at room temperature and liquid when heated, it can act as a cold barrier in the system. However, M1000 contains large numbers of C and O elements that reduce the thermal stability of *α*-ZrP. Therefore, based on our previous work [20], adding both phosphated cellulose (PCF) and exfoliated *α*-ZrP synergistically increases the flame retardancy of PVA.

The directional freezing method, also known as an ice template method, is a commonly used method for preparing aerogels. It is generally employed in biological tissue engineering, thermal insulation, refractory materials, and lightweight high-strength materials. Ice crystals are removed by freeze-drying to obtain porous materials with orientated structures using orientated ice crystals as templates during the freeze-drying process [21]. At the same time, adjusting the rigid part of the matrix or the flexible interface to obtain new micro and nano materials [22] combined with directional freezing can result in multi-functional aerogel materials. The aerogels prepared by the directional freezing method possess an anisotropic porous structure and exhibit better insulation behavior by preventing local heat concentration and increasing the overall heat transfer barrier [23]. Directional freezing has only one temperature gradient, while multidirectional freezing has two or three temperature gradients simultaneously [24]. Zhao et al. [25] fabricated an elastic network of chitosan aerogels by a novel bidirectional freeze-casting method, which provided combined advantages of super-elasticity and flame retardancy along with highly oriented wavy-layered stacking microstructures. It is generally employed in biological tissue engineering, thermal insulation, refractory materials, and lightweight high-strength materials [26,27,28].

In this study, a lightweight, multi-anisotropic aerogel based on PVA and two kinds of nanofillers (PCF and exfoliated *α*-ZrP) were prepared through a multi-directional freezing technique. *α*-ZrP with layered structures and nanofibrous PCF have a synergistic effect on PVA aerogels, by acting as structural support and improving the thermal insulation of the aerogel. Comparison of the aerogels prepared by unidirectional freezing and bidirectional freezing showed that the composite aerogel prepared through the multi-directional freezing technique possessed a denser lamellar porous structure, which improves heat diffusion in-plane and enhances cold insulation. Hence, they are potentially useful as flexible thermal insulators for both residential and commercial construction.

## 2. Results and Discussion

### 2.1. Structural and Morphological Analyses of PCF

The digital photographs of *α*-cellulose and aqueous PCF suspensions are shown in Figure 1. Both these compounds could be dispersed in water after stirring magnetically within 10 s. After 24 h, the PCF remained as a light-yellow-colored suspension, whereas clear sedimentation of *α*-cellulose occurred, which demonstrated the excellent dispersibility and stability of PCF in water. FTIR spectroscopy was used to characterize the chemical bonds in *α*-cellulose and PCF, as shown in Figure 1a,b. The spectrum of *α*-cellulose showed two peaks at 3350 and 2990 cm^−1^, corresponding to O-H and C-H stretching vibrations, respectively. The peak at 1210 cm^−1^ in the spectrum of PCF indicated the presence of P=O in the structure, due to the replacement of primary hydroxyl group with the phosphate group. The three peaks at 1028 cm^−1^, 993 cm^−1^, and 900 cm^−1^ were due to P-OH and P-O-C bonds of PCF. Thermogravimetric analysis (TGA) was conducted to determine the thermal stabilities of *α*-cellulose and PCF. The TGA curves of both *α*-cellulose and PCF showed one-stage weight loss patterns (Figure 1c). For *α*-cellulose, the main degradation occurred between 300 and 420 °C, which corresponded to the rupture of *α*-cellulose macromolecular chains, with the release of small molecular substances such as CO, CO_2_, and H_2_O. Meanwhile, the main degradation occurred between 150 and 230 °C, which could be ascribed to the decomposition of H_3_PO_4_ grafted onto the PCF. The DTG curves of *α*-cellulose and PCF are presented in Figure 1d. PCF showed the maximum rate of mass loss at 212.4 °C, whereas for *α*-cellulose it was between 330 and 340 °C. Meanwhile, the entire mass loss of *α*-cellulose was much higher than that of PCF, proving that PCF had remarkable thermal stability. In particular, the initial decomposition temperature of *α*-cellulose was 212.5 °C, which was more than that of PCF (183.7 °C). This indicated that the introduction of phosphoric acid groups promoted the thermal degradation of *α*-cellulose. The TGA data is presented in Appendix A.

### 2.2. Chemical Structure and Morphological Analysis of α-ZrP

To prove the successful exfoliation of *α*-ZrP, a series of qualitative and quantitative tests and characterizations were conducted, and the results are shown in Figure 2. Figure 2a,b are the digital photographs showing the influence of 24 h of deposition on the appearance of primordial and exfoliated *α*-ZrP suspensions in water with 0.4 g/mL *α*-ZrP solids content. The exfoliated *α*-ZrP solution clearly retained its stable appearance without noticeable sedimentation, but sediments appeared in the primal *α*-ZrP sample after the storage period. Figure 2c shows the XRD patterns of *α*-ZrP. The four characteristic peaks centered at 11.7°, 19.7°, 24.9°, and 34.0° were indexed to the (002), (110), (112), and (312) planes of primal *α*-ZrP, respectively (JCPDS 33-1482). The peak intensity at 11.7° corresponded to an interlayer distance of 0.76 nm [29,30]. Compared to the primal *α*-ZrP, all peak intensities of the exfoliated *α*-ZrP weakened, widened, and became offset, which was attributed to the peeling effect of M1000 on *α*-ZrP. The micro-morphology of *α*-ZrP was studied by TEM (Figure 2d). Obviously, in the method that used M1000, the peeling effect resulted in lamellar *α*-ZrP. This demonstrated that the exfoliated *α*-ZrP had a regular hexagon structure and the size of the nanosheet layer was about 100 nm. The thermal stability of *α*-ZrP in nitrogen atmosphere was evaluated by TGA (Figure 2e,f). Both samples exhibited one-stage weight-loss patterns. The pyrolysis of primal *α*-ZrP mainly occurred between 500 and 600 °C, which corresponded to the condensation of P-OH to Zr(HPO_4_)_2_, with the removal of water of crystallization, and finally ZrP_2_O_7_ was obtained [31]. Meanwhile, in the case of exfoliated *α*-ZrP, the main degradation occurred between 280 and 400 °C, which was due to the decomposition of M1000, intercalated between the *α*-ZrP nanosheets. The specific data is presented in Appendix A. These results show that *α*-ZrP was successfully exfoliated.

### 2.3. Discussion on the Unidirectional PVA Aerogels

#### 2.3.1. Formation and Structure of the Unidirectional PVA Aerogels

Generally, the micro-structure of composite aerogels are influenced by the amount of nanofillers [32]. SEM investigated the effect of nanofillers on the morphological changes of the composite aerogels at the cross-sections and along the length. The fracture morphologies of pure PVA aerogel in Figure 3a,b show a 3D network structure with hollow interiors and extremely uniform holes. Meanwhile, a highly ordered pore structure parallel to the growth plane of the ice crystal was also observed. Even after loading with PCF and *α*-ZrP, the PVA/PCF/*α*-ZrP composite aerogels retained the layered porous structure without uniform voids and obvious collapse (Figure 3c–l). This result could be due to two reasons: on one hand, the introduced PCF and *α*-ZrP showed improved dispersion in the polymer matrix, which prevented the collapse of its structure. On the other hand, the strong adhesivity of PVA increased the skeletal strength [33]. Figure 3c,d show the cross-sectional and longitudinal SEM images of PVA/*α*-ZrP_20_, respectively. Obviously, with the addition of *α*-ZrP into PVA, the pores at the cross section of the composite aerogel were broken, but the shape of the aerogel along its length became uniform and dense. Contrastingly, the composite aerogel containing only 20% PCF showed a layered structure regardless of the direction, but this structure was neither regular nor close-compacted (Figure 3k) and even showed faults (Figure 3l). The SEM image of PVA/PCF_5_/*α*-ZrP_15_ is shown in Figure 3e,f. With PCF to *α*-ZrP ratio of 1:3, the gaps between the aerogel layers appeared filled, and the entire layer displayed a uniform and regular morphology. However, the edge of the layered structure at the cross section was jagged and the edges of the longitudinal section were curled. Subsequently, with the increase in the ratio of PCF to *α*-ZrP, a strong and thick layered structure began to appear in the aerogel, as shown in Figure 3g,h. It was evident that with the increase in the nanofiller proportion, the wall thickness and strength gradually became more obvious and serious. However, for PVA/PCF_15_/*α*-ZrP_5_ (Figure 3i,j), the composite aerogel showed broken cell walls. This shows that a further increase in the ratio of PCF to *α*-ZrP had a bad effect on the microstructure of the aerogel. In summary, PCF to *α*-ZrP ratio of 1:1 provided excellent structure for the composite aerogels.

The energy dispersive spectrometry (EDS) intuitively and clearly analyzed the elemental compositions and contents of each composite. The distribution and contents of C, O, P, and Zr elements in PVA/PCF_10_/*α*-ZrP_10_ composite aerogel are shown in Figure 4b–f. Obviously, each element inhabited in its own area, symbolizing that the PCF and *α*-ZrP components were relatively evenly scattered in the aerogel. The energy spectrum and elemental distribution of other composite aerogels are shown in Appendix A. Similar to PVA/PCF_10_/*α*-ZrP_10_, PVA/PCF_x_/*α*-ZrP_y_ composite aerogel in Appendix A showed an obvious and uniform elemental distribution, once again proving that the two nanofillers could be consummately dispersed in the PVA matrix. This point was also proven by XRD results of PVA aerogel and PVA/PCF_10_/*α*-ZrP_10_ aerogel (Appendix A).

#### 2.3.2. Mechanical Properties of the Unidirectional PVA Aerogels

The effects of PCF and *α*-ZrP nanofillers on the mechanical and compression performance can be seen from the stress-strain curves and Young’s moduli in Figure 5 and Appendix A. The values shown are the average of five measurements of each sample under the same conditions. Appendix A shows the compression performance along the direction parallel to the growth of ice crystal of pure PVA and PVA/*α*-ZrP_20_ composite aerogel. PVA aerogel has many structural defects and so it yields when its compressive strain reaches 68 wt%. The compressive strain of PVA/*α*-ZrP20 composite aerogel improved slightly to 70%, due to the formation of an effective long strip microstructure (Appendix A). When the ratio of PCF to *α*-ZrP was changed from 1:3 to 1:1 and then to 3:1, the compressive stress was obviously improved to 5.56, 5.47, and 4.96 MPa, respectively. This indicated that the change in the proportion of nanofillers had little effect on the mechanical properties of the aerogel in the longitudinal direction. Meanwhile, with the incorporation of 20.0 wt% of pure PCF into PVA aerogels, its tensile strength was 4.74 MPa. The differences in their layered structure in the PVA aerogels (Figure 3b,d,f,h,j,l) resulted in uneven forces, which were prone to breakage when compressed.

The Young’s moduli and specific moduli of pure PVA and PVA composite aerogels are shown in Figure 5c. Different from the results of compressive stress, except for PVA/PCF_10_/*α*-ZrP_10_, the Young’s moduli of other composite aerogels with nanofillers were reduced to varying extents, which could be due to the presence of thin and curled or misaligned holed-walls in the vertical direction. The increase in specific modulus could be due to the fact that the addition of nanofillers markedly reduced the density of the PVA composite aerogels. The compressive stress, Young’s modulus, and specific modulus of PVA/PCF_10_/*α*-ZrP_10_ were 5.47 MPa, 0.5 MPa, and 8.9 m^2^/s^2^, respectively, indicating that the 1:1 ratio of PCF and *α*-ZrP endowed the PVA aerogel with outstanding performance. The digital photos of longitudinally compressed PVA/PCF_10_/*α*-ZrP_10_ aerogel are shown in Appendix A. The uniformity in the expansion of composite aerogel indicated that the pressure at all points was balanced. This could also be observed in the SEM image (Figure 3h). Next, the stress-strain curves perpendicular to the direction of ice crystal growth were studied. Similar to the results of longitudinal stress-strain curve, the PVA aerogel showed a small stress of 0.29 MPa ascribed to its ice crystal direction. PVA aerogel has a small Young’s modulus of 0.14 MPa ascribed to its larger aperture. When the addition amount of PCF was increased from 0 wt% to 20 wt%, the addition amount of *α*-ZrP reduced from 20 wt% to 0 wt%, and the stress dramatically changed from 0.09 MPa to 0.82 MPa. The digital photos of horizontally compressed PVA/PCF_10_/*α*-ZrP_10_ aerogel are shown in Appendix A.

#### 2.3.3. Thermal Stabilities of Unidirectional PVA Aerogels

The thermal stabilities of PVA aerogel and its nano-composite aerogels in nitrogen and air were evaluated by TGA, as shown in Figure 6. In Figure 6a,b, the PVA aerogel and its nano-composite aerogels showed similar two-stage thermal degradation patterns in both environments. As expected, when 20 wt% of PCF and *α*-ZrP together were introduced, the residual carbon contents of PVA nanocomposite aerogels at 800 °C improved from 3.98% (pure PVA, N_2_) and 2.84% (pure PVA, air) to 24.2%, 14.48% (N_2_) and 9.33%, 8.65% (air). Due to the high thermal stability of PCF and low thermal stability of exfoliated *α*-ZrP, the residual carbon contents of PVA/PCF_x_/*α*-ZrP_y_ were higher than PVA/*α*-ZrP_20_ and lower than PVA/PCF_20_, which was consistent with the results in Figure 1c and Figure 2e.

In the case of both air and nitrogen atmospheres, the two stages of decomposition of PVA aerogels corresponded to the scission and cyclization of PVA chains and the decomposition of polyene [34]. It was evident that the initial degradation temperatures (T_5%_, Appendix A) and the first maximum degradation temperatures (T_max1_, Appendix A) of the PVA nano-composite aerogels under both conditions were lower than those of pure PVA aerogel, which could be ascribed to the degradation of PCF and M1000 in exfoliated *α*-ZrP. Furthermore, PCF and M1000 started breaking, accompanied by the formation of polyphosphoric acid (PPA) along with the release of ammonia and water. However, the second maximum degradation temperatures (T_max2_) for both air and nitrogen atmospheres of the PVA/*α*-ZrP_20_ were still lower than the pure PVA aerogel, which could be associated with the loading of *α*-ZrP.

When two or more additives perform significantly better than each individual case, a synergistic effect index (SE) can be used to mathematically and quantitatively describe its relationship [35]. The SE of PCF and *α*-ZrP are defined by Equation (1):(1)$$\mathrm{SE}\;=\;\frac{E_{PVA+xPCF+y\left(\alpha -ZrP\right)}-E_{PVA}}{\frac{x}{x+y}\left(E_{PVA+\left(x+y\right)PCF}-E_{PVA}\right)+\frac{y}{x+y}\left(E_{PVA+\left(x+y\right)(\alpha -ZrP)}-E_{PVA}\right)}$$
where *x* indicates the amount of PCF and *y* the amount of *α*-ZrP in PVA/PCF_x_/*α*-ZrP_y_ composite aerogels. An SE value of 1 stipulates superposition, a value >1 indicates synergism and value below <1 indicates antagonism. According to Equation (1), synergism occurs only if there is an additional advantageous interaction between PCF and *α*-ZrP. Therefore, the synergistic effect index of the initial decomposition temperature and the residual carbon amount at 800 °C was calculated using Equation (1) and presented in Table 1. For any composite aerogel, the synergy index greater than 1 indicated that PCF and *α*-ZrP had an effect of 1 + 1 and greater than 2 in composite aerogel.

#### 2.3.4. Thermal Insulation Properties of the Unidirectional PVA Aerogels

To determine the effectiveness of nanofillers in thermal management, an infrared camera combined with an alcohol lamp was assembled to detect the temperatures on the upper surfaces of composite aerogels, as shown in Figure 7. The top surfaces of the PVA/PCF_x_/*α*-ZrP_y_ composites remained at relatively low temperatures (Figure 7e–j) after being heated by an alcohol lamp for 120 s. Compared to pure PVA aerogel, the surface temperatures of PVA composite aerogels were significantly lower. When the samples were heated for 120 s, the surface temperatures of PVA/*α*-Zr_20_, PVA/PCF_5_/*α*-ZrP_15_, PVA/PCF_10_/*α*-ZrP_10_, PVA/PCF_15_/*α*-ZrP_5_, and PVA/PCF_20_ aerogels were 46.87 °C, 48.03 °C, 45.98 °C, 46.47 °C, and 51.56 °C, respectively (Figure 7b). In order to further verify the insulation performance of the material at low temperature, the PVA/PCF_10_/*α*-ZrP_10_ aerogel was placed directly on the steel plate on a heat source at 60 °C. After 300 s, the surface temperature of the aerogel increased from 12.13 °C to 21.3 °C and remained at this temperature, which indicated that PVA/PCF_10_/*α*-ZrP_10_ composites had good thermal insulation property.

In short, it was evident that PVA/PCF_10_/*α*-ZrP_10_ aerogel was the best choice in terms of mechanical properties, thermal stability, and microstructure. Hence, PVA/PCF_10_/*α*-ZrP_10_ was chosen as the matrix for multi-directional freezing.

### 2.4. Discussion of the Multi-Directional PVA/PCF_10_/α-ZrP_10_-x Aerogels

#### 2.4.1. Formation and Structure of the Multi-Directional PVA/PCF_10_/*α*-ZrP_10_-x Aerogels

The microstructures of PVA/PCF_10_/*α*-ZrP_10_ in different freezing directions can be seen in Figure 8. For ease of illustration, we define the X-axis as radial direction and the Y-axis and Z-axis as axial directions. During the multi-directional freezing of PVA/PCF_10_/*α*-ZrP_10_ suspension, the temperature gradients in different directions generate ice crystals that nucleate and grow at different rates into parallel lamellae. As expected, the PVA/PCF_10_/*α*-ZrP_10_-x aerogels show a layered structure with spacings between adjacent lamellae that vary in the range of 5–100 μm. It was observed that the freezing directions of aerogels greatly affect the lamellar structure along the XY plane, especially the bilateral freezing (Figure 8d) and multilateral freezing (Figure 8g). The reason could be that the XY surface of the mold was exposed to low-temperature ethanol, whereas the XZ and YZ planes were exposed to air. Hence, the propagation of temperature was slower than that of XY plane, which hindered the unidirectional growth of ice crystal and formed a denser and broken layer. PVA/PCF_10_/*α*-ZrP_10_-x aerogels showed a neat parallel arrangement of lamellar structure at both X-Z and Y-Z cross-sections in the SEM images and the wall thickness was determined to be 3 to 10 μm by multi-directional freezing technique. As compared to PVA/PCF_10_/*α*-ZrP_10_-1 (Figure 8b,c) and PVA/PCF_10_/*α*-ZrP_10_-2 (Figure 8e,f), the bilateral lamellar structure of PVA/PCF_10_/*α*-ZrP_10_-3 (Figure 8h,i) was more uniform and dense. The wall surface of PVA/PCF_10_/*α*-ZrP_10_-3 (Figure 8g–i) with holes was the roughest with nanofibers bulging between the lamellae. This confirmed the presence of PCF nanofibers and the average dispersion of *α*-ZrP in PVA matrix. This anisotropic structure of PVA/PCF_10_/*α*-ZrP_10_-x aerogels proved the effectiveness of multi-directional freezing technique, which favored thermal insulation. Elemental mapping of PVA/PCF_10_/*α*-ZrP_10_-x aerogels (Appendix A) indicted good proportional distribution of C, O, P, and Zr elements, thus corroborating the successful synthesis of PVA/PCF_10_/*α*-ZrP_10_-x.

#### 2.4.2. Mechanical Properties of Multi-Directional PVA/PCF_10_/*α*-ZrP_10_-x Aerogels

The multi-directional PVA/PCF_10_/*α*-ZrP_10_-x aerogels presented a unique anisotropic mechanical performance due to their highly ordered microstructures. As shown in Figure 9a–c, the compression performance of the aerogels was excellent only in the vertical direction (XY), but poor in other directions (XZ and YZ). There was 80% compression in the direction perpendicular to the lamellae (Figure 9a), PVA/PCF_10_/*α*-ZrP_10_-3 (8.25 MPa) displaying stronger compressive stress compared with PVA/PCF_10_/*α*-ZrP_10_-2 (6.76 MPa) and PVA/PCF_10_/*α*-ZrP_10_-1 (5.66 MPa). Similarly, in Figure 9b,c, PVA/PCF_10_/*α*-ZrP_10_-3 still showed excellent compression resistance in the other two directions. This was due to the influence of the support and interpenetration of the three-dimensional aerogel layers based on the multi-directional freezing mode.

The Young’s moduli and specific moduli of PVA-based composite aerogels are summarized in Figure 9d–f. The densities of the PVA/PCF_10_/*α*-ZrP_10_-1, PVA/PCF_10_/*α*-ZrP_10_-2, and PVA/PCF_10_/*α*-ZrP_10_-3 aerogels were 0.058 g/cm^3^, 0.056 g/cm^3^, and 0.05 g/cm^3^, respectively. It is worth noting that the density of PVA/PCF_10_/*α*-ZrP_10_-3 aerogel was distinctly reduced, which could be due to the formation of tiny pores in the cell wall (Figure 8g–i). The compressive modulus and specific modulus of PVA/PCF_10_/*α*-ZrP_10_-3 increased dramatically along different directions. In the XY plane, the Young’s modulus and specific modulus of PVA/PCF_10_/*α*-ZrP_10_-1 were 0.3 MPa and 5.2 m^2^/s^2^, respectively. In case of bidirectional freezing, the Young’s modulus and specific modulus of PVA/PCF_10_/*α*-ZrP_10_-2 aerogel increased dramatically to 0.46 MPa and 8.2 m^2^/s^2^ and these were higher than those of the PVA/PCF_10_/*α*-ZrP_10_-1, which underwent multi-directional freezing. The Young’s modulus and specific modulus of PVA/PCF_10_/*α*-ZrP_10_-3 aerogel increased to 0.72 MPa and 14.4 m^2^/s^2^, an increase of 2.4 and 2.77 times, respectively, as compared to those of the PVA/PCF_10_/*α*-ZrP_10_-1 aerogel. The increasing trends of Young’s moduli and specific moduli for PVA/PCF_10_/*α*-ZrP_10_-1, PVA/PCF_10_/*α*-ZrP_10_-2, and PVA/PCF_10_/*α*-ZrP_10_-3 were all similar in the XZ and YZ planes. This could be ascribed to the incorporation of PCF and *α*-ZrP into the network and also the porous interpenetration caused by different freezing methods, which provided more points to withstand the pressure PVA/PCF_10_/*α*-ZrP_10_-x aerogels. These results implied that the compressive properties of PVA/PCF_10_/*α*-ZrP_10_-x aerogels were considerably enhanced after multi-directional freezing. Appendix A shows the comparison between the density and compressive stress applied in this work with other modified PVA-based aerogels [9,10,12,36,37,38]. The findings showed that the nanofillers and freezing methods had a positive effect on the density and mechanical properties of composite aerogels.

### 2.5. Thermal Insulation Properties of Aerogels

Figure 10a shows the effectiveness of multi-directional aerogels in thermal management. Compared to PVA/PCF_10_/*α*-ZrP_10_-1 aerogel (Figure 7h), the temperatures on the upper surfaces of PVA/PCF_10_/*α*-ZrP_10_-2 and PVA/PCF_10_/*α*-ZrP_10_-3 did not change significantly on heating with alcohol lamp. The temperatures were 45.1 °C and 43.2 °C, respectively, which indicated that changing the freezing method had a limited effect on the thermal insulation performance. The cold insulation performances of PVA and PVA/PCF_10_/*α*-ZrP_10_-x were also confirmed, as shown in Figure 10b, when the sample was placed on the cold source at −20 °C for five minutes. The surface temperature of PVA aerogel was 9.4 °C, the temperature at the center for PVA/PCF_10_/*α*-ZrP_10_-1 was 11.2 °C, whereas those of PVA/PCF_10_/*α*-ZrP_10_-2 and PVA/PCF_10_/*α*-ZrP_10_-3 were 11.8 °C and 13.6 °C, respectively. Obviously, PVA/PCF_10_/*α*-ZrP_10_-3 showed more efficient cold insulation efficiency than PVA, PVA/PCF_10_/*α*-ZrP_10_-1, and PVA/PCF_10_/*α*-ZrP_10_-2. The two reasons could be as follows. On one hand, the added M1000 stripping *α*-ZrP could fill the voids in PVA/PCF_10_/*α*-ZrP_10_-x aerogel and add to the defects of PVA/PCF_10_/*α*-ZrP_10_-x aerogel, which was solid at room temperature and impeded the formation of efficient cold conduction paths. On the other hand, the multi-directional freezing method increased the probability of contact and collisions between the nanofillers and PVA matrix and formed a more complex grid path, which could consume part of the coldness, and reduce the cold transfer rate.

### 2.6. Combustion Behaviors of Aerogels

The cone calorimeter test (CCT) is a comprehensive and authoritative technique to evaluate the flame-retardancy of materials, which are closer to the real burning behaviors [39,40]. Curves for heat release rate (HRR) of PVA, PVA/PCF_10_/*α*-ZrP_10_-1, PVA/PCF_10_/*α*-ZrP_10_-2, and PVA/PCF_10_/*α*-ZrP_10_-3 aerogels are shown in Figure 11a. Pure PVA showed a peak of heat release rate (PHRR) at 314.72 Kw/m^2^, when ignited for 12 s (TTI, Figure 11f). With addition of 10 wt% PCF and 10 wt% *α*-ZrP into PVA, the PHRR of PVA/PCF_10_/*α*-ZrP_10_-1 (Figure 11e, labeled as P-1) was dramatically reduced to 223.6 kW/m^2^. The PHRRs of bidirectional PVA/PCF_10_/*α*-ZrP_10_-2 (Figure 11e, labeled as P-2) and multidirectional PVA/PCF_10_/*α*-ZrP_10_-3 (Figure 11e, labeled as P-3) were 217.72 kW/m^2^ and 217 kW/m^2^, respectively. As expected, the HRR and PHRR of PVA/PCF_10_/*α*-ZrP_10_-3 were delayed, but the ignition time (TTI, Figure 11f, labeled as P-3) increased. Similar results were also obtained from the THR curves in Figure 11b. Obviously, pure PVA had a high THR of 13.13 MJ/m^2^. The incorporation of nanofillers and changing the freezing method caused a significant decrease in THR values of PVA/PCF_10_/*α*-ZrP_10_-1, PVA/PCF_10_/*α*-ZrP_10_-2, and PVA/PCF_10_/*α*-ZrP_10_-3 to 9.58 MJ/m^2^, 9.55 MJ/m^2^, and 8.73 MJ/m^2^, respectively. All the above results attested that the addition of PCF and *α*-ZrP in PVA, combined with multi-directional freezing, could successfully delay the PHRR and THR of PVA composites during the burning process.

It has been reported that over 60% of deaths in fire accidents are caused due to inhalation of toxic smoke. Figure 11c,d show the amount of CO generated per second (COP) and total smoke production (TSP) of PVA and its composites. Compared to pure PVA, the peak COP and TSP values of PVA/PCF_10_/*α*-ZrP_10_-3 decreased to 0.121 g/s and 0.003 m^2^, respectively, corresponding to a decrease of 42.7% and 74.6%, respectively. The reason for the sharp reduction in THR value of PVA/PCF_10_/*α*-ZrP_10_-3 was the addition of PCF and zirconium phosphate. However, the maximum possibility was that the layered interlaced complex structure formed by multi-directional freezing brought about cross-linking of the aerogel during combustion and inhibited the release and production of smoke.

The fire hazards for PVA composites were also predicted by means of fire performance index (FPI) and fire growth index (FGI), which are defined by the ratios of TTI to PHRR and PHRR to the time to PHRR (TPHRR), respectively. As shown in Figure 11f, pure PVA showed a low FPI value of 0.032 m^2^·s/kW and a high FGI value of 5.43 kW/m^2^·s. In comparison, PVA/PCF_10_/*α*-ZrP_10_-x composite aerogels (PVA/PCF_10_/*α*-ZrP_10_-1:P-1; PVA/PCF_10_/*α*-ZrP_10_-2:P-2; and PVA/PCF_10_/*α*-ZrP_10_-3:P-3) exhibited much higher FPI and lower FGI values. In particular, PVA/PCF_10_/*α*-ZrP_10_-3(P-3) showed the highest FPI value of 0.685 m^2^·s/kW and the lowest FGI value of 3.91 kW/m^2^·s.

The limiting oxygen index (LOI) and UL-94 vertical burning tests were conducted to estimate the flame retardancy of PVA composite aerogels, as shown in Figure 11g,h. The LOI of PVA was 18.1%, which indicated that it is highly flammable. In case of PVA/PCF_10_/*α*-ZrP_10_-x composite aerogels, the LOI significantly improved to more than 28.4%, which suggested that PCF and *α*-ZrP had a stimulating effect on the flame retardancy of PVA. Simultaneously, the results of UL-94 vertical burning tests showed that pure PVA had no rating (NR), due to the presence of large numbers of carbon, hydrogen, and oxygen elements. The combustion grade of PVA/PCF_10_/*α*-ZrP_10_-x aerogel was V0, due to the presence of large amount of P element and a small amount of N of M1000, thus proving its good flame retardancy. Moreover, PVA aerogel (Appendix A) continued to burn after heating for 10 s with the flame of an alcohol lamp. In sharp contrast to this, the PVA/PCF_10_/*α*-ZrP_10_-x composites (Appendix A) showed no obvious traces of burning and self-extinguished after the flame was removed.

### 2.7. Mechanism of Flame Retardancy

#### 2.7.1. Vapor Phase Analysis

Coupled thermogravimetric analysis-infrared spectrometry (TG-IR) was employed to determine the thermal decomposition behaviors and distinguish the volatile components of PVA and PVA/PCF_10_/*α*-ZrP_10_-x composite aerogels during pyrolysis under N_2_ atmosphere [41,42]. Appendix A shows the three-dimensional diagrams and FT-IR spectra versus temperature obtained by TG-IR. Pure PVA was primarily decomposed into water vapor (3750 cm^−1^), alkane (2940 cm^−1^), CO_2_ (2360 cm^−1^), and CO (2190 cm^−1^) in IR spectra. However, PVA/PCF_10_/*α*-ZrP_10_-x behaved differently. Compared to the pure PVA aerogel, a higher number of pyrolytic products of PVA/PCF_10_/*α*-ZrP_10_-x ascertained the formation of carbonyl compounds (1710 cm^−1^), P-O (1210 cm^−1^), P=O (568 cm^−1^), due to the decomposition of PCF and the presence of the P element. Moreover, the peak at 1510 cm^−1^ was attributed to aromatic compounds, which suggested extensive carbonization and conversion of polysaccharides in PCF to aromatic graphitized structures (char) during pyrolysis [43,44]. Curves for the intensities of common gaseous products are plotted in Figure 12, wherein the four types could be divided into two categories: non-flammable volatiles (CO_2_, water vapor) and flammable volatiles (CO, alkyl compounds). According to the Lambert–Beer law, the concentration of gas linearly depends on the absorption intensity [45]. The maximum absorption intensity for water vapor of PVA/PCF_10_/*α*-ZrP_10_-x was significantly improved compared with pure PVA. This showed that the addition of P improved the thermal decomposition of composite materials and a small amount of M1000 could volatilize the amine-containing compounds. More particularly, the maximum absorbance intensities of gaseous products (CO, CO_2_) and flammable C-H compounds escaping from PVA/PCF_10_/*α*-ZrP_10_-x were significantly reduced in comparison to pristine PVA. This demonstrated the eminent suppressive effect of PCF and *α*-ZrP on the release of volatiles in the pyrolysis of PVA.

#### 2.7.2. Condensed Phase Analysis

The micro-morphology and graphitization degree of residual chars after combustion have also been studied by SEM, EDS, and Raman spectroscopy to further interpret the mechanism of flame-retardancy. The SEM images of residual carbons from PVA, PVA/PCF_10_/*α*-ZrP_10_-1, PVA/PCF_10_/*α*-ZrP_10_-2, and PVA/PCF_10_/*α*-ZrP_10_-3 are presented in Figure 13a–d. The residual carbon in PVA contained a large number of fine needle-like structures (Figure 13a), which could be attributed to an orderly microstructure achieved by unidirectional freezing. Therefore, PVA could not hinder the transfer of heat and combustible gases, since it could not act as a complete barrier and would collapse during the combustion process. The residues of PVA/PCF_10_/*α*-ZrP_10_-x prepared by different freezing methods showed a lamellar microstructure formed by the stacking of *α*-ZrP nano-plates. The condensed residues served as a non-flammable physical barrier with excellent catalytic carbonization properties that protected the underlying aerogels, due to Zr^4+^ frames in the microstructure. Furthermore, compared to PVA the addition of PCF and *α*-ZrP flakes integrated the morphology of residual carbon in PVA/PCF_10_/*α*-ZrP_10_-1, but had still few micropores (Figure 13b). PVA/PCF_10_/*α*-ZrP_10_-2 showed a gradual denser char layer, due to the dual barrier effects of nanofillers and bilateral freezing. At the same time, the carbon layer of PVA/PCF_10_/*α*-ZrP_10_-3 (Figure 13d) was more complete and dense, benefitting from the same effect.

EDX (Appendix A) was used to determine the chemical compositions and elemental distributions of the burning residues for exploring the flame-retardancy property. From PVA (Appendix A) to PVA/PCF_10_/*α*-ZrP_10_-1 (Appendix A), the carbon fraction (wt%) of residue increased dramatically as PCF and *α*-ZrP were incorporated into the aerogel. This indicated that the degradation products of the aerogel were effectively converted into carbon, as a result of the strong catalytic effect of PCF and *α*-ZrP. With the change in the freezing method, the residual carbon content of the composite aerogel was slightly increased (Appendix A). This indicated that the multi-directional freezing method was hindered with the escaping of gas. In addition, Appendix A also proves the enrichment and even distribution of P and Zr elements, which further improved the thermal stability of aerogels during the burning process.

Raman spectroscopy is widely used for the analysis of carbon layers. Figure 14 shows the Raman spectra of carbon layers of composite materials PVA, PVA/PCF_10_/*α*-ZrP_10_-1, PVA/PCF_10_/*α*-ZrP_10_-2, and PVA/PCF_10_/*α*-ZrP_10_-3 after cone calorimetry. Two broad diffraction peaks were observed. The one at approximately 1360 cm^−1^ (D band) corresponded to the first-order Raman scattering of sp^3^ hybridized C atoms. This showed defects in the graphitic layer and indicated a disordered carbon structure. The other peak at approximately 1580 cm^−1^ was the G band and it corresponded to the sp^2^ hybridized carbon atoms in the graphitic layer. This was suggestive of e_2g_-type vibrations in the lattice network of the graphitic layer and indicated an ordered carbon structure. Since the D band and G band in the spectrum overlapped, the Gaussian peak fitting was employed to deconvolute it into two individual peaks. The degree of graphitization of carbonaceous materials was expressed by the ratio of the intensity of D peak area (I_D_) and the intensity of G peak area (I_G_) [46,47]. Commonly, a lower I_D_/I_G_ value indicated that the carbon layer contained more graphitic-type carbon, which led to the perfect quality of the carbon layer and an excellent barrier performance. Amazingly for PVA, its I_D_/I_G_ value was up to 7.26 (Figure 14a). After the incorporation of 10 wt% PCF and 10 wt% *α*-ZrP into PVA, the I_D_/I_G_ value dropped sharply to 5.83 (Figure 14b). This was due to the combined effect of the excellent catalytic ability of *α*-ZrP for carbon formation along with a rich carbon source of PCF. Meanwhile, the small amount of M1000 provided the combustible gas and PVA/PCF_10_/*α*-ZrP_10_-1 became a comprehensibly efficient flame-retardant system. Subsequently, the I_D_/I_G_ values of PVA/PCF_10_/*α*-ZrP_10_-2 and PVA/PCF_10_/*α*-ZrP_10_-3 continuously decreased to 3.96 and 3.11, respectively. Since the bilateral freezing and multilateral freezing randomly complicated the microscopic pore structures of the materials, improved the crosslinking of the products after combustion, and decreased the defects, the I_D_/I_G_ value decreased.

#### 2.7.3. Proposed Mechanism of Flame Retardancy

Based on the above results, a plausible mechanism of flame retardancy of PVA/PCF_10_/*α*-ZrP_10_-3 was proposed to explain the synergistic effects between *α*-ZrP and PCF in the PVA aerogels. As shown in Figure 15, the mechanism of flame-retardancy could be explained considering two aspects: vapor phase flame retardancy and condensed phase flame retardancy [48]. Phosphate groups at the C6 hydroxyl group on PCF surfaces underwent dephosphorylation to generate aromatic graphitized structures (char) and phosphate units. These phosphate units on PCF underwent dehydration and condensation to form polyphosphates of PCF (Reaction 1). Moreover, the pyrolysis also caused cracking of PCF into P•, which captured H• and OH• free radicals. This diluted the oxygen concentration in air, reduced the number of combustibles, and thus hampered the conduction of combustion [49]. The existing phosphate groups also catalyzed the dehydration of PVA chains and promoted the development of thermally stable char structures [20]. Meanwhile, M1000 used for the exfoliation of *α*-ZrP was pyrolyzed into non-combustible gases, like ammonia and nitrogen, which reduced the number of combustible gases and further suffocated the flame.

Moreover, *α*-ZrP, which is a layered acidic activator and furnishes Brønsted (H^+^) and Lewis (Zr^4+^) acidic sites on its surface, and plays a synergistic role by initiating the formation of residues and consolidates the stability of the carbonaceous layer through catalytic reactions [50]. Therefore, the PVA chains could be catalytically carbonized by the Brønsted (H^+^) acid during the burning process. Meanwhile, the degradation products were captured by the Lewis (Zr^4+^) acid to form a dense and substantial number of residues, due to the presence of Zr^4+^ which facilitated dehydrogenation, crosslinking, cyclization, condensation, and aromatization reactions (Reaction 2) [51]. It was easy to catalyze the aerogel prepared by multi-directional freezing to obtain a carbon layer, due to its uniform and dense microstructure, which reduced the escaping of gas. Finally, the continuous char layers could efficiently reduce the heat exchange between PVA and air, the free radicals could effectively dilute the air, and their mutual co-operation could achieve the flame retardancy of PVA.

## 3. Materials and Methods

### 3.1. Materials

Phosphite 98%, urea 99.5%, phosphoric acid 85%, zirconyl chloride (ZrOCl_2_·8H_2_O), 98%, PVA (PVA-1799), anhydrous ethanol, and acetone were purchased from Chengdu Kelon Chemical Reagent Factory (Sichuan, China). *α*-Cellulose (25 μm length) was supplied by Innochem (Beijing, China). A commercial polyoxyalkylene amine, (H_3_C(OCH_2_CH_2_)_19_(OCH_2_CHCH_3_)_3_NH_2_, Jeffamine, reported average molecular weight of 1,000) was procured from Deji Trading Co., Ltd., Shanghai, China. A dialysis bag with a molecular weight cut-off of 2000 was used for the experiments. All the chemicals were used as received without further purification.

### 3.2. Fabrication

#### 3.2.1. Preparation of Phosphated Cellulose

PCF was synthesized according to a method reported in the literature [52] and also described in a previous article [20]. The reaction mechanism for the synthesis of PCF is shown in Appendix A. Briefly, urea (12.48 g; 0.208 mol) was added to a 250 mL three-neck flask and heated to 140 °C; it was held at this temperature for 30 min under a N_2_ atmosphere until a fully melted state was observed. Then, *α*-cellulose (2 g; 0.0123 mol) and phosphite (10.28 g; 0.1253 mol) were alternately added to the molten urea in sequence to reduce foaming. The reaction was allowed to proceed at 150 °C for 5 h for full response to the final formation of pale-yellow solids. The product was dissolved in 100 mL of deionized water, and then aqueous ethanol was added; the mixture was manually stirred to precipitate the sediment. This process was repeated three times to remove residual urea and unreacted phosphite. After filtration, the final product (PCF) was obtained via drying the precipitate ethanol solution.

#### 3.2.2. Preparation and Exfoliation *α*-ZrP

*α*-ZrP was prepared by using a reflux method in a previously reported procedure [53] and the specific process is as follows. ZrOCl_2_·8H_2_O (20 g) was refluxed with 200 mL 3.0 M H_3_PO_4_ in a single-necked flask at room temperature for 60 min and then held at 100 °C for 24 h. After the completion of the reaction, the products were washed with deionized water till pH was 7 and dried at 65 °C for 24 h. The dried *α*-ZrP product was ground using a mortar and pestle into fine powder.

*α*-ZrP (0.5 g) was weighed and dispersed in 25 mL acetone by sonication for 1 h. M1000 solution (2.77 mL, 0.6 g/mL) was slowly added drop-wise to the stirring *α*-ZrP dispersion. The reaction mixture was stirred for 4 h, then sonicated for 60 min, and rotated the acetone out using a rotary evaporator. This was followed by ultrasonic dispersion in deionized water and centrifugation at 10,000 rpm for 30 min in polytetrafluoroethylene tubes. The precipitate was removed and the relatively clear suspension contained only exfoliated *α*-ZrP and excess of M1000. The clear suspension was taken in a dialysis bag and dialyzed for 48 h to remove excess polyoxyalkylene amine. The mass fraction of exfoliated *α*-ZrP was 0.04 g/mL. The reaction mechanism for *α*-ZrP synthesis is shown in Appendix A.

#### 3.2.3. Fabrication of the Unidirectional PVA Aerogel

Different mass ratios of PCF/*α*-ZrP(1:3, 1:1, and 3:1) were added in the sample bottle to obtain stable suspensions. The total flame-retardant component of the PVA aerogel was maintained at 20 wt% (Appendix A). The steps for the preparation of PVA/PCF_10_/*α*-ZrP_10_ are as follows: firstly, PVA (0.8 g) and PCF/*α*-ZrP (1:1) dispersion (0.2 g) were added to 20 mL deionized water and stirred magnetically at 100 °C for 3 h to obtain a uniform solution. The mixture was poured into a mold with a copper plate as the bottom surface, which was kept in contact with an ethanol bath (−50 °C) to prepare unidirectional PVA aerogel. Similar steps were also followed for the syntheses of PVA/*α*-ZrP_20_, PVA/PCF_5_/*α*-ZrP_15_, PVA/PCF1_5_/*α*-ZrP_5_, PVA/PCF_20_ aerogels. Finally, the unidirectional PVA aerogels were freeze-dried at a sublimating temperature of −47 °C and pressure of 1 Pa for 5 days. The process for the preparation of unidirectional PVA aerogel is shown in Appendix A.

#### 3.2.4. Fabrication of the Multi-Directional PVA Aerogel

The aqueous dispersion of the PVA composite was transferred to a mold with different freezing directions, as shown in Appendix A. Thus, the aqueous PVA solution generated different temperature gradients along each direction. The anisotropic aerogel for two- or three-dimensional orientation was obtained by freeze-drying under the same conditions as above. The final aerogels were labeled as PVA/PCF/*α*-ZrP-x (PVA/PCF/*α*-ZrP-1, PVA/PCF/*α*-ZrP-2, PVA/PCF/*α*-ZrP-3) according to the number of dimensions.

### 3.3. Measurements and Characterization

The structures of pulverized PCF powder samples were characterized by Fourier transform infrared spectroscopy (FTIR, Nicolet 6700 in the spectral range of 4000 to 400 cm^−1^ at a resolution of 4 cm^−1^.

X-ray diffractometry (XRD) was conducted using X’Pert PRO at a scanning rate of 10°/min in the 2θ range of 10° to 60° and the excitation source with Cu K*α* radiation (*λ* = 0.154 nm).

Micromorphologies of the samples were studied using scanning electron microscope (SEM), equipped with a JSM-7500F, at an accelerating voltage of 15.0 kV. The samples were sprayed with gold prior to analyses. The elements and their distribution were evaluated by energy dispersive spectrometry (EDS).

Transmission electron microscopy (TEM, FEI Tecnai F20) was employed to inspect the exquisite morphology of exfoliated *α*-ZrP.

The mechanical properties of samples were determined using a universal testing machine (CMT4104), wherein the samples were compressed with a specific strain of 80% at a rate of 1 mm/min.

Thermogravimetric analysis (DSC823 TGA/SDTA85/e) was conducted to determine the thermal stabilities of the samples in the temperature range from 40 °C to 800 °C at a heating rate of 20 °C/min in nitrogen and air atmospheres.

Experiments on the combustion of aerogels were conducted to investigate the combustion behaviors of aerogel samples. The limiting oxygen index (LOI) values were determined using a JF-3 Oxygen Index Flammability Gauge (Jiangning, China), according to GB/T 2046.2–2009. The dimensions of all the samples were 100 mm × 10 mm × 10 mm. Vertical burning tests (UL-94) were performed using CZF-4 instrument (Jiangning, China), according to GB/T 8333-2008. The dimensions of all the samples were 100 mm × 10 mm × 10 mm. Cone calorimetry was done using an ASTME1354/ISO 5660 model to investigate the fire performance of polymer materials, which were wrapped in an aluminum foil and the testing was done under conditions of 35 kW/m^2^ heat flow. The dimensions of the specimens were 100 mm × 100 mm × 10 mm.

Thermogravimetric analysis–Fourier transform infrared (TG-FTIR) spectroscopic analysis was performed using a Perkin–Elmer STA 6000 model, under nitrogen and air atmospheres, at a linear heating rate of 20 °C/min from 40 °C to 800 °C.

The Raman spectral analysis was conducted on a Laser micro-Raman spectrometer (IM 52) in the spectral range of 400 to 1800 cm^−1^.

## 4. Conclusions

PVA composite aerogels with excellent flame retardancy and thermal insulation properties were successfully prepared through a multi-directional freezing method by using a combination of PCF and *α*-ZrP as nanofillers. This multi-directional freezing method not only effectively optimized the microstructure of a composite aerogel, but also promoted cross-linking, thus significantly improving its thermal stability. Particularly, the infrared thermal-imaging picture showed thermal insulation, cold insulation performance, and the outstanding flame retardancy of the aerogel. It confirmed an effective synergy between PCF and *α*-ZrP and the formation of pathways that hindered energy exchange. With the incorporation of 10 wt% PCF and 10 wt% *α*-ZrP, the peak heat release rate (PHRR), total smoke release (TSR), and CO production (COP) of PVA composite fibers were considerably decreased by 42.3%, 41.4%, and 34.7%, respectively. The LOI value and UL-94 grade test results showed that the composite aerogel met the standards of flame retardancy. We expect that the multi-directional freezing method, combined with the synergistic property in the making of composite materials, would provide new avenues for the manufacturing of composite aerogels with remarkable flame retardancy for applications in building materials.

## Figures and Tables

**Figure 1 ijms-23-15919-f001:**
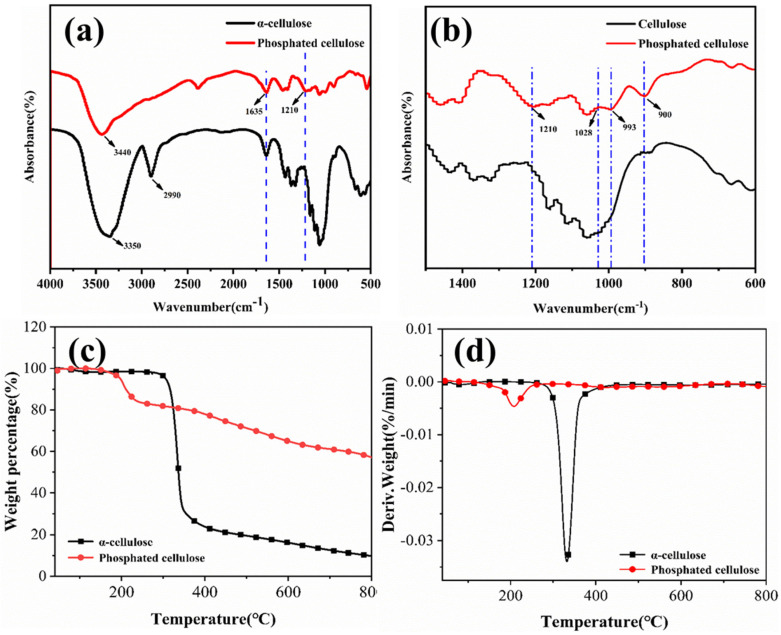
Analyses of *α*-cellulose and PCF: (**a**,**b**) FTIR spectra; (**c**) TGA curves; and (**d**) DTG curves.

**Figure 2 ijms-23-15919-f002:**
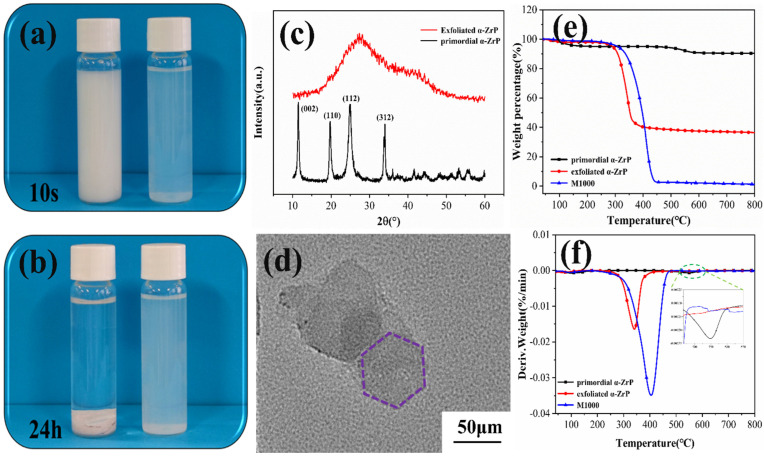
Analyses of exfoliated *α*-ZrP and primordial *α*-ZrP: (**a**,**b**) photographs of the suspensions; (**c**) Raman spectra; (**d**) TEM image of exfoliated *α*-ZrP; (**e**) TGA curves; and (**f**) DTG curves.

**Figure 3 ijms-23-15919-f003:**
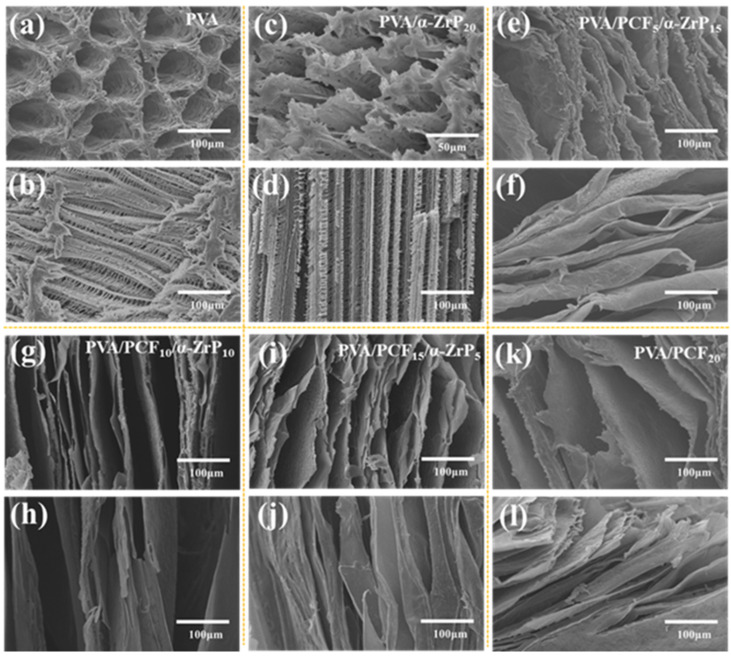
SEM images of the cross sectional and longitudinal sections of PVA/PCF_x_/*α*-ZrP_y_ aerogels. (**a**,**b**): PVA; (**c**,**d**): PVA/*α*-ZrP_20_; (**e**,**f**): PVA/PCF_5_/*α*-ZrP_15_; (**g**,**h**): PVA/PCF_10_/*α*-ZrP_10_; (**i**,**j**): PVA/PCF_15_/*α*-ZrP_5_; and (**k**,**l**): PVA/PCF_20_.

**Figure 4 ijms-23-15919-f004:**
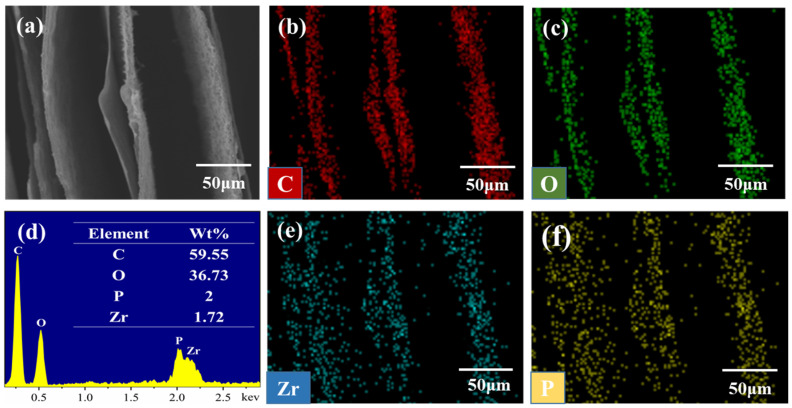
(**a**) SEM image; elemental mapping of PVA/PCF_10_/*α*-ZrP_10_: (**b**) carbon (C), (**c**) oxygen (O), (**d**) EDS spectra and content, (**e**) zirconium (Zr), and (**f**) phosphorus (P).

**Figure 5 ijms-23-15919-f005:**
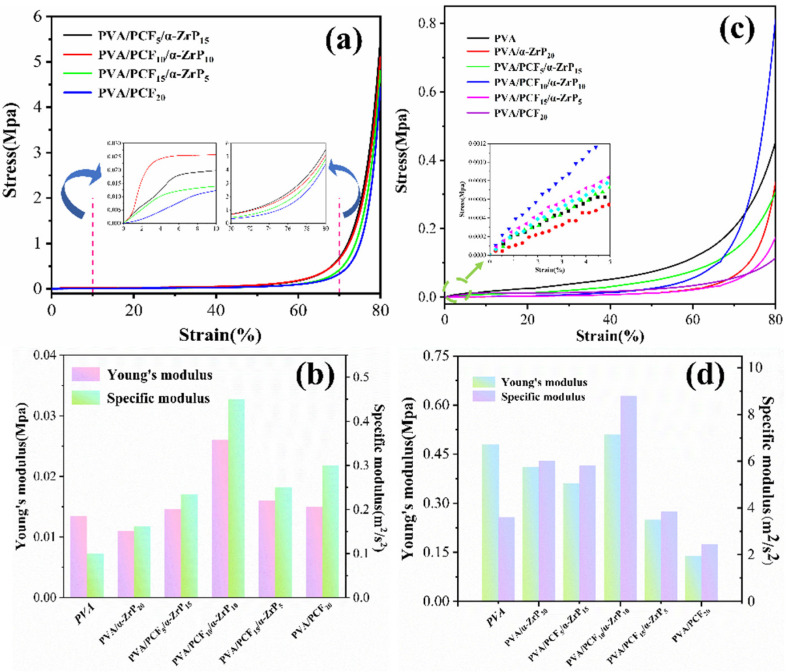
(**a**) vertically compressed stress-strain curves; (**b**) horizontally compressed stress-strain curves; (**c**) vertical Young’s modulus and specific modulus; (**d**) horizontal Young’s modulus and specific modulus.

**Figure 6 ijms-23-15919-f006:**
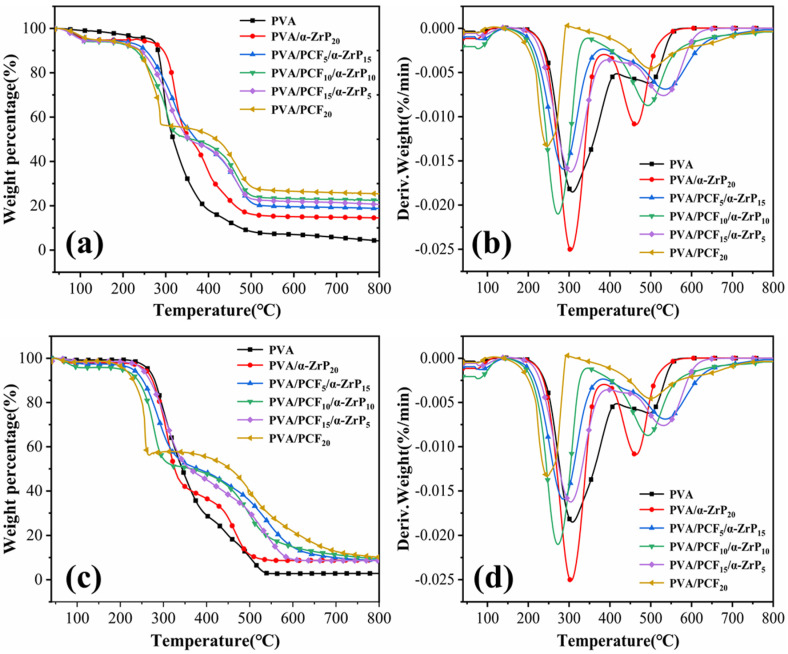
TGA and DTG curves of PVA/PCF_x_/*α*-ZrP_y_ aerogel in (**a**,**b**) nitrogen atmosphere and (**c**,**d**) air atmosphere.

**Figure 7 ijms-23-15919-f007:**
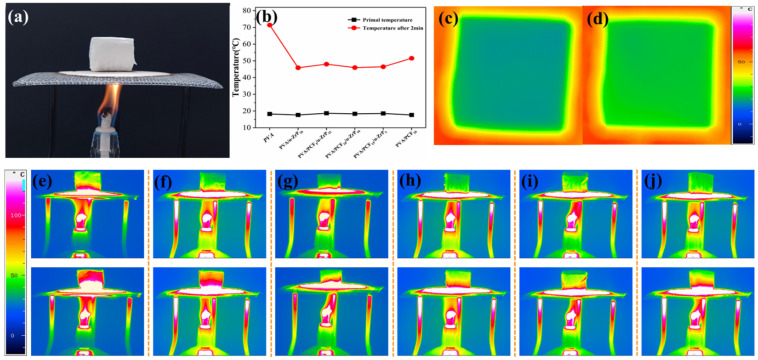
Application of thermal management of PVA/PCF_x_/*α*-ZrP_y_. (**a**) Photograph of the heating process; (**b**) Comparison of the original temperature of aerogel and the temperature after heating for 2 min; (**c**,**d**) Infrared thermal imaging of PVA/PCF_10_/*α*-Zr_10_ in 60 °C after 5 min. Thermographic images of PVA (**e**), PVA/*α*-Zr_20_ (**f**), PVA/PCF_5_/*α*-ZrP_15_ (**g**), PVA/PCF_10_/*α*-ZrP_10_ (**h**), PVA/PCF_15_/*α*-ZrP_5_ (**i**), and PVA/PCF_20_ (**j**) aerogels on a stage heated by alcohol lamp fire.

**Figure 8 ijms-23-15919-f008:**
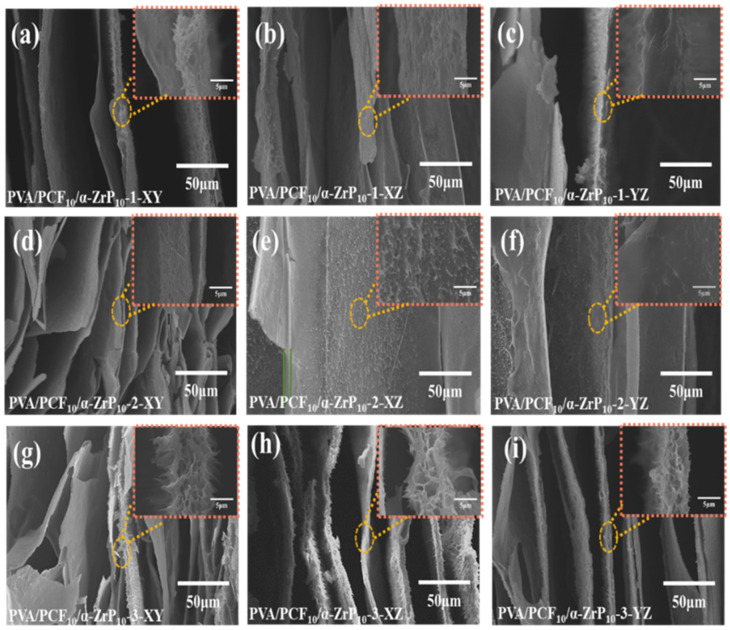
SEM images for the multi-directional PVA/PCF_10_/*α*-ZrP_10_-x aerogels. (**a**) PVA/PCF_10_/*α*-ZrP_10_-1-XY; (**b**) PVA/PCF_10_/*α*-ZrP_10_-1-XZ; (**c**) PVA/PCF_10_/*α*-ZrP_10_-1-YZ; (**d**) PVA/PCF_10_/*α*-ZrP_10_-2-XY; (**e**) PVA/PCF_10_/*α*-ZrP_10_-2-XZ; (**f**) PVA/PCF_10_/*α*-ZrP_10_-2-YZ; (**g**) PVA/PCF_10_/*α*-ZrP_10_-3-XY; (**h**) PVA/PCF_10_/*α*-ZrP_10_-3-XZ; and (**i**) PVA/PCF_10_/*α*-ZrP_10_-3-YZ.

**Figure 9 ijms-23-15919-f009:**
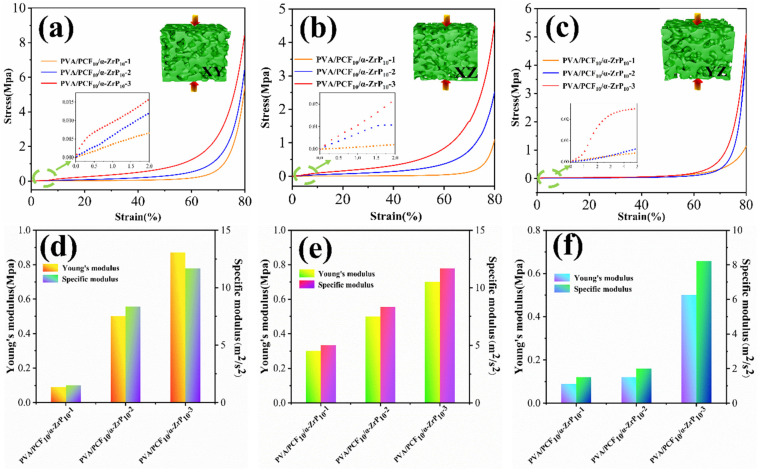
Compressive properties of the multi-directional PVA/PCF_10_/*α*-ZrP_10_-x aerogels. (**a**–**c**) Stress–strain curves in different directions; (**d**–**f**) Young’s moduli and specific moduli in different directions.

**Figure 10 ijms-23-15919-f010:**
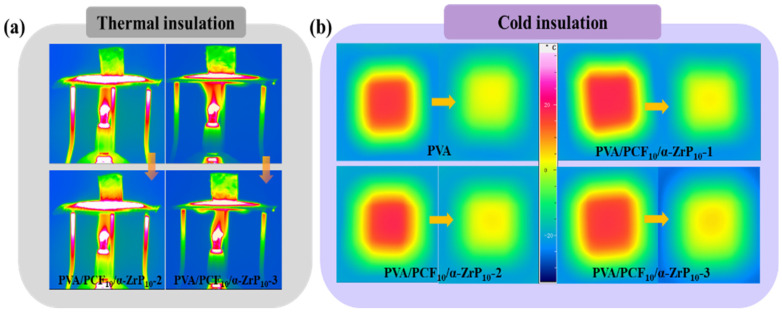
(**a**) Thermal insulation properties of PVA/PCF_10_/*α*-ZrP_10_-2 and PVA/PCF_10_/*α*-ZrP_10_-3 aerogels; (**b**) Cold insulation properties of PVA and PVA/PCF_10_/*α*-ZrP_10_-x aerogels.

**Figure 11 ijms-23-15919-f011:**
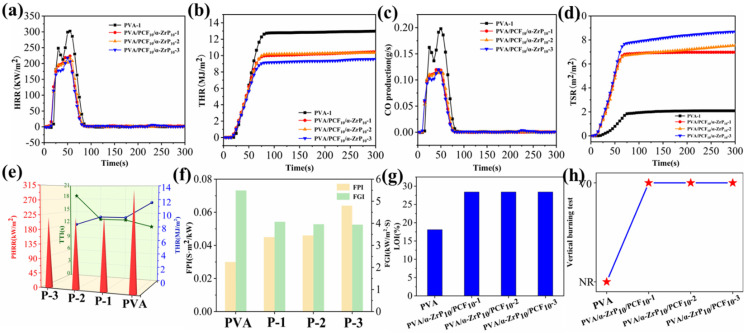
(**a**) HRR curves, (**b**) THR curves, (**c**) COP curves, (**d**) TSP curves, (**e**) PHRR, TTI, and THR values, (**f**) FPI and FGI values, (**g**) LOI values, and (**h**) UL-94 vertical testing of PVA and PVA/PCF_10_/*α*-ZrP_10_-x composite aerogels.

**Figure 12 ijms-23-15919-f012:**
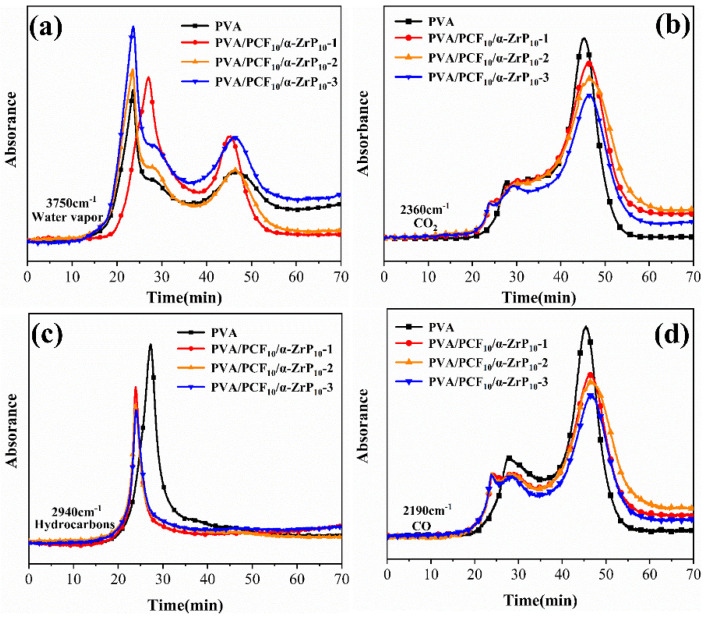
Absorbances of pyrolysis products of PVA and PVA/PCF_10_/*α*-ZrP_10_-x vs. time: (**a**) water vapor, (**b**) CO_2_, (**c**) hydrocarbons, and (**d**) CO.

**Figure 13 ijms-23-15919-f013:**
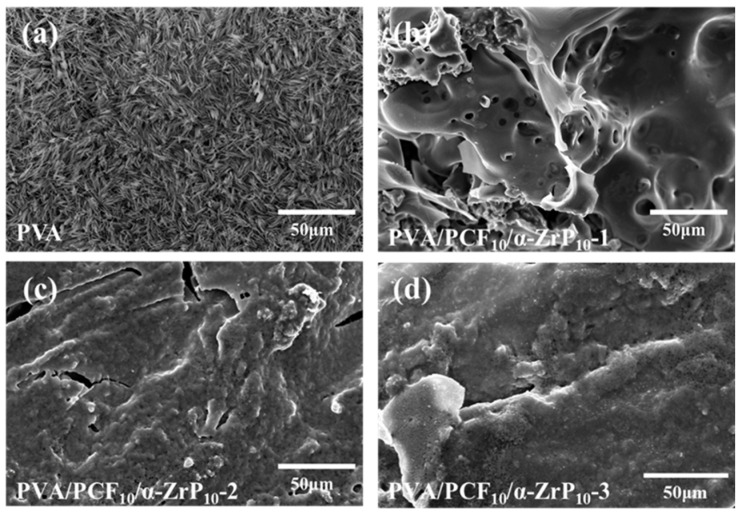
SEM images of the residues of (**a**) PVA, (**b**) PVA/PCF_10_/*α*-ZrP_10_-1, (**c**) PVA/PCF_10_/*α*-ZrP_10_-2, and (**d**) PVA/PCF_10_/*α*-ZrP_10_-3 after cone calorimetry.

**Figure 14 ijms-23-15919-f014:**
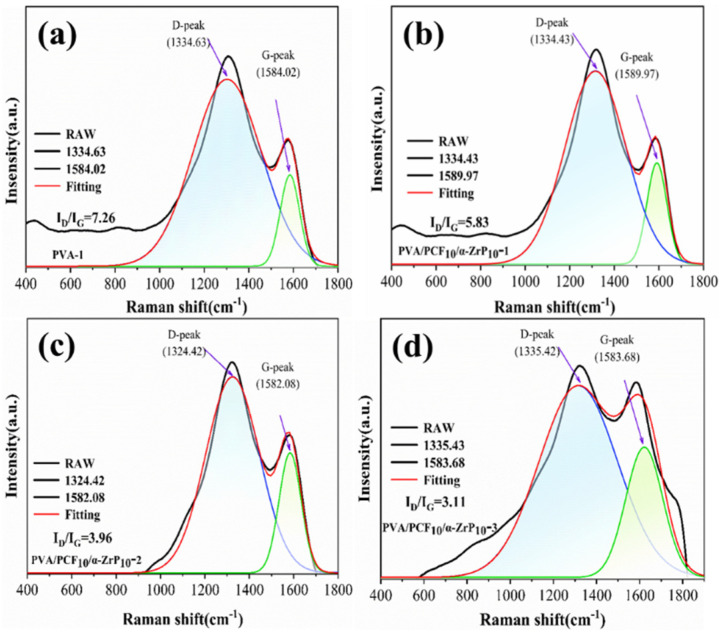
Raman spectra of the residues: (**a**) PVA, (**b**) PVA/PCF_10_/*α*-ZrP_10_-1, (**c**) PVA/PCF_10_/*α*-ZrP_10_-2, and (**d**) PVA/PCF_10_/*α*-ZrP_10_-3.

**Figure 15 ijms-23-15919-f015:**
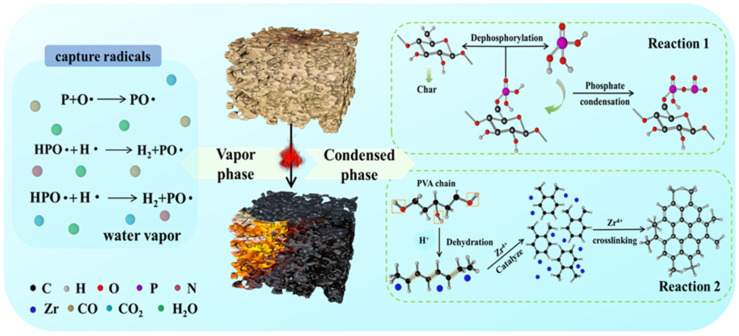
Mechanism of flame-retardancy of PVA/PCF_10_/*α*-ZrP_10_-3.

**Table 1 ijms-23-15919-t001:** SE values of aerogel with TGA data.

Sample	SE			
	*T* _donset_		Char Residues	
	N_2_	Air	N_2_	Air
PVA/PCF_5_/*α*-ZrP_15_	1.14	1.02	1.009	2.03
PVA/PCF_10_/*α*-ZrP_10_	1.39	1.13	1.13	2.33
PVA/PC_15_/*α*-ZrP_5_	1.06	1.03	1.14	1.39

## Data Availability

Not applicable.

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
