# Peer review of "Preparation, Properties, and Mechanism of Flame-Retardant Poly(vinyl alcohol) Aerogels Based on the Multi-Directional Freezing Method"

_ijms, 2022, doi:10.3390/ijms232415919_

Round 1

Reviewer 1 Report

This manuscript discussed a method for fabricating PVA aerogels with flame retardant property. I see multiple effort on testing and characterization for the aerogels and also some research potential in industry &academia. However this manuscript seems a little lengthy and could be improved in multiple areas. I would suggest authors to address these questions first.

(1) This manuscript is submitted to ijms, but the authors seem not explain enough the relation to molecular science in abstract and introduction, or it is only a materials/polymer paper. This is actually critical and can be improved.

(2) This work has 19 figures and 24 pages actually some parts could be put into supplemental. Is figure 16-19 all necessary in main text? Authors cannot just put figures there (like fig.16) and wrote "The reaction mechanism for the synthesis of PCF is shown in Fig. 16." Fig.17  Also, Fig. 16(a)(b) is not professional. Figure 19 is not clear.

(3) It is a preparation/properties based work, so I personally would suggest to put Materials and Methods chapter in front of results. 

(4)If multi-directional is critical in this work, introduction needs to explain the concept and difference with "directional","multi-directional".

(5) It is easily to become out of focus when dealing with multi-directional freezing part and flame-retardant part, maybe can put some into supplemental to focus on the multi-directional freezing.

Reviewer 2 Report

This is an excellent and rather extensive study, and both methods and results are presented rather well, just not in the usual order. I would strongly suggest to simply swap chapters 2 and 3. On top of that, I only have some minor comments:

- Zirconium phosphate is also used in non-polar polymers as flame retardant, where the mechanism should be different. This should be discussed in the in the introduction.

- The "directional freezing method" should be explained better.

- In Figure 11, the diagrams are at the border of readability, and either it should be split into two figures or at least the legends of a-d and the x-axis labels of g & h should be bigger. The same holds for Figures 9 and 14.

Round 2

Reviewer 1 Report

The revised manuscript has answered the reviewer's all questions thoroughly and I would suggest to publish in present form.